# Clonal Expansion of Tumor-Infiltrating T Cells and Analysis of the Tumor Microenvironment within Esophageal Squamous Cell Carcinoma Relapsed after Definitive Chemoradiation Therapy

**DOI:** 10.3390/ijms22031098

**Published:** 2021-01-22

**Authors:** Takahiro Mori, Kenichi Kumagai, Keisuke Nasu, Takamasa Yoshizawa, Koji Kuwano, Yoshiki Hamada, Hideki Kanazawa, Ryuji Suzuki

**Affiliations:** 1Departments of Clinical Oncology and Gastroenterological Surgery, National Hospital Organization Sagamihara National Hospital, 18-1 Sakuradai, Minami-ku, Sagamihara 252-0392, Japan; 2Department of Rheumatology and Clinical Immunology, Clinical Research Center for Rheumatology and Allergy, National Hospital Organization Sagamihara National Hospital, 18-1 Sakuradai, Minami-ku, Sagamihara 252-0385, Japan; kumagaik-ora@h.u-tokyo.ac.jp (K.K.); pd19008@stu.tsurumi-u.ac.jp (K.N.); pd19010@stu.tsurumi-u.ac.jp (T.Y.); kuwano.koji.gq@mail.hosp.go.jp (K.K.); suzuki@repertoire.co.jp (R.S.); 3Department of Oral and Maxillofacial Surgery, School of Dental Medicine, Tsurumi University, 2-1-3 Tsurumi, Tsurumi-ku, Yokohama 230-8501, Japan; hamada-y@tsurumi-u.ac.jp; 4Department of Surgery, National Hospital Organization Sagamihara National Hospital, 18-1 Sakuradai, Minami-ku, Sagamihara 252-0392, Japan; kanazawa.hideki.ma@mail.hosp.go.jp; 5Tohoku University Graduate School of Medicine, 1-1 Seiryo-machi, Aobaku, Sendai 980-8574, Japan; 6Department of Oral-Maxillofacial Surgery and Orthodontics, The University of Tokyo Hospital, 7-3-1 Hongo, Bunkyo-ku, Tokyo 113-8655, Japan

**Keywords:** esophageal squamous cell carcinoma, definitive chemoradiation therapy, tumor-infiltrating T cell, TCR repertoire, PD-L1, CD8, salvage esophagectomy, prognostic marker

## Abstract

(1) Background: Comparable prognoses after definitive chemoradiation therapy (CRT) to surgery alone for esophageal squamous cell carcinoma (ESCC) have been previously reported; however, no robust prognostic markers have been established. The clonality of tumor-infiltrating lymphocytes (TILs) and tumor microenvironments (TMEs) in ESCC relapsed after CRT were examined to explore prognostic markers. (2) Methods: Clonality of TIL and TME were examined in ESCC with and without preceding CRT, as well as oral squamous cell carcinoma (OSCC) and healthy volunteers as controls. The clonality of TIL was assessed by T-cell receptor (TCR) α and β repertoire analyses and evaluated by diversity indices. The TME was assessed by quantitative polymerase chain reaction evaluating *PD-L1* and *CD8B*. (3) Results: The clonal expansion of TIL was significantly induced within ESCCs and OSCCs, when compared to healthy volunteers, and was mostly induced within ESCCs after definitive CRT. Diversity indices of TIL were not associated with the prognosis, but the ratio of *PD-L1* mRNA to *CD8B* mRNA in TME was significantly associated with a poor prognosis after salvage surgery (*p* = 0.007). (4) Conclusions: The clonal expansion of TIL is induced after definitive CRT for ESCC, and the ratio of *PD-L1* mRNA to *CD8B* mRNA within tumor tissues is a prognostic marker candidate for salvage esophagectomy after CRT.

## 1. Introduction

Esophageal squamous cell carcinoma (ESCC) is one of the most lethal neoplasms in both Japan and other parts of the world [1,2,3]. Multimodal medical treatments consisting of neoadjuvant chemotherapy or chemoradiation therapy followed by surgery is the current gold standard of treatment, especially for patients with advanced stage II or III ESCCs [4,5,6]. Alternatively, definitive chemoradiation therapy (CRT) itself may be another powerful treatment for carcinoma curability, because some cases have been reported to achieve pathologically complete remission (pCR) after neoadjuvant CRT [7]. Comparable prognoses after definitive CRT to surgery alone have been reported by a couple of independent prospective studies [8,9]. Nevertheless, not all patients with ESCC have been treated with definitive CRT as a first-line treatment. One of the reasons to refrain from definitive CRT as a first-line treatment option is that local relapses may then require salvage esophagectomies, which may have a high risk for severe comorbidities such as pulmonary complications [10]. Hence, it is necessary to treat them using the optimal medical procedures that have been determined by robust predictive or prognostic markers. Various studies have been performed to identify the predictive or prognostic factors of medical procedures for ESCC, including CRT, although no robust marker has yet been established [11,12,13].

Recent advances in molecular biology and tumor immunology have revealed the fact that the therapeutic effects of chemotherapy or radiation should be greatly influenced by the tumor immunologic microenvironment [14,15,16,17]. Actually, many studies have suggested that the factors relating to the tumor microenvironment (TME), such as PD-L1 expression or the presence of tumor-infiltrating lymphocytes (TIL), could be prognostic or predictive marker candidates for chemotherapy or chemoradiation in various types of human cancers, including ESCC [18,19,20,21,22,23]. These previous reports have encouraged us to examine and characterize the specific features of TIL and TME, especially within ESCCs that have relapsed after definitive CRT, to explore the prognostic maker candidates of definitive CRT followed by salvage esophagectomy. However, almost all the patients with advanced ESCCs are treated with neoadjuvant CRT or chemotherapy before surgery in Japan [24,25]. This suggests the difficulty of enrolling a sufficient number of patients with advanced ESCCs that have been solely treated by surgery as a control group. Instead, we enrolled patients with oral squamous cell carcinoma (OSCC) in this study; OSCC is considered to be comparable to ESCC, because they share similar pathologic diagnoses, squamous cell carcinoma, carcinogenic pathways (such as alcohol drinking and smoking), and molecular features [26,27,28,29]. Additionally, ESCC is usually treated by neoadjuvant CRT or chemotherapy followed by surgery, as described above, whereas OSCC is rarely treated with neoadjuvant chemotherapy or CRT before surgery.

In this study, we reveal the results of diversity analyses of the T-cell receptor (TCR) repertoire of TIL within post-definitive CRT locally relapsed ESCC by comparing them to ESCC and OSCC that have both been treated only by surgery in order to reveal the specific immunological features initiated by definitive CRT. We also demonstrate the molecular features of the TME, especially in ESCCs relapsed after definitive CRT, which could be prognostic marker candidates after salvage esophagectomy.

## 2. Results

### 2.1. Clonal Expansion of Tumor-Infiltrating T Lymphocytes Was Significantly Induced within Esophageal and Oral Squamous Cell Carcinoma Tumor Tissues and Mostly Induced within Esophageal Squamous Cell Carcinoma That Relapsed after Definitive CRT

The results of the repertoire analyses of complementarily determining region three of *TCR α* (*TRA*) or *β* (*TRB*) are shown in three dimensional graphs in Figure 1. The diversity and evenness of the TCR repertoire are defined as the Shannon-Weaver index H, inverse Simpson’s index (1/λ), and Pielou’s evenness in Table 1 or Appendix A. Based on the results of the Shapiro-Wilk *W* test and Levene’s test, the mean value of each clinical group was statistically compared to an unpaired Student’s *t*-test, Welch’s *t*-test, or Mann-Whitney U test, as described below (Figure 2).

The obtained results indicated that the divergency and evenness of the TCR repertoire in healthy volunteers were always significantly higher than in patients with OSCC (*p* < 0.001 in the Shannon-Weaver Index H for *TRA*, *p* < 0.001 in Pielou’s evenness for *TRA*, *p* < 0.001 in inverse Simpson (1/λ) for *TRA*, *p* < 0.001 in the Shannon-Weaver Index H for *TRB*, *p* < 0.001 in Pielou’s evenness for *TRB*, and *p* = 0.002 in inverse Simpson (1/λ) for *TRB*); in patients with ESCC treated by surgery alone (*p* < 0.001 in the Shannon-Weaver Index H for *TRA*, *p* < 0.001 in Pielou’s evenness for *TRA*, and *p* = 0.001 in the Shannon-Weaver Index H for *TRB*); or in patients with ESCC that relapsed after definitive CRT (*p* < 0.001 in the Shannon-Weaver Index H for *TRA*, *p* < 0.001 in Pielou’s evenness for *TRA*, *p* < 0.001 in the Shannon-Weaver Index H for *TRB*, and *p* < 0.001 in Pielou’s evenness for *TRB*), except, in some cases, as follows: inverse Simpson (1/λ) for *TRA* either in patients with ESCC surgery alone (*p* = 0.444) or with ESCC recurrence after definitive CRT (*p* = 0.182), inverse Simpson (1/λ) for *TRB* either in patients with ESCC surgery alone (*p* =0.491) or with ESCC recurrence after definitive CRT (*p* = 0.236), and Pielou’s evenness for *TRB* for patients with ESCC surgery alone (*p* = 0.236) (Figure 2). It should be noted that the results in healthy volunteers were based on the different materials, such as peripheral white blood cells.

As for OSCCs and ESCCs, RNAs were extracted from tumor tissues that were obtained by surgical resection and used for TCR sequencing. No statistically significant differences were observed between patients with OSCC and ESCC; both treated by surgery alone (*p* = 0.288 in the Shannon-Weaver index H for *TRA*, *p* = 0.582 in Pielou’s evenness for *TRA*, *p* = 0.800 in inverse Simpson (1/λ) for *TRA*, *p* = 0.501 in the Shannon-Weaver index H for *TRB*, *p* = 0.538 in Pielou’s evenness for *TRB*, and *p* = 0.800 in inverse Simpson (1/λ) for *TRB*) and the mean value of each diversity index was similar between these two groups (Figure 2). On the other hand, statistically significant differences were observed between the OSCC group and ESCC that relapsed after definitive CRT (*p* < 0.001 in the Shannon-Weaver index H for *TRA*, *p* < 0.001 in Pielou’s evenness for *TRA*, *p* < 0.001 in the Shannon-Weaver index H for *TRB*, *p* < 0.001 in Pielou’s evenness for *TRB*, and *p* = 0.014 in inverse Simpson (1/λ) for *TRB*), except for inverse Simpson (1/λ) for *TRA* (*p* = 0.387) (Figure 2). Between the ESCC treated by surgery alone group and ESCC relapsed after definitive CRT group, the differences were not statistically significant, although the mean diversity indices were always higher in those of surgery alone than in those that relapsed after definitive CRT. This was likely due to the limited number of patients with ESCC treated by surgery alone (*p* = 0.252 in the Shannon-Weaver index H for *TRA*, *p* = 0.171 in Pielou’s evenness for *TRA*, *p* = 0.418 for inverse Simpson (1/λ) for *TRA*, *p* = 0.518 in the Shannon-Weaver index H for *TRB*, *p* = 0.525 for inverse Simpson (1/λ) for *TRB*, and *p* = 0.460 in Pielou’s evenness for *TRB*).

### 2.2. Divergency of the T-Cell Receptor Repertoire of Tumor-Infiltrating Lymphocytes Was Not Significantly Different between Patients with Good or Poor Prognoses after Salvage Surgery for Esophageal Squamous Cell Carcinoma Locally Relapsed after Definitive Chemoradiation Therapy

The results of the divergency evaluation for the TCR repertoire of TIL for locally relapsed ESCCs after definitive CRT are shown in Figure 3A,B. The divergency was not statistically different between patients with good prognoses and those with poor prognoses (*p* = 0.765 for the Shannon-Weaver index H, *p* = 0.525 for inverse Simpson (1/λ), and *p* = 0.958 for Pielou’s evenness for *TRA* and *p* = 0.599 for the Shannon-Weaver Index H, *p* = 0.631 for inverse Simpson (1/λ), and *p* = 0.668 for Pielou’s evenness for *TRB*).

### 2.3. Ratio of PD-L1 mRNA to CD8B mRNA within Tumor Tissues Significantly Higher in Patients with Poor Prognoses after Salvage Surgery for Locally Relapsed Esophageal Squamous Cell Carcinoma after Definitive Chemoradiation Therapy

The results of the quantitative PCR (qPCR) analyses are shown in Figure 4. The ratio of *PD-L1* mRNA to *CD8B* mRNA was statistically different between patients with good prognoses and with poor prognoses after the salvage surgeries for ESCCs relapsed after definitive CRT (*p* = 0.007), although the expression levels of *PD-L1* and *CD8B*, adjusted by *GAPDH* mRNA, were not statistically different between these two groups (*p* = 0.311 and *p* = 0.490, respectively).

## 3. Discussion

To our knowledge, this is the first report to analyze the clonality of the TIL and TMEs within ESCCs that relapsed after definitive CRT. Our results suggest that clonal expansion of the TIL was induced after definitive CRT; moreover, the ratio of *PD-L1* mRNA to *CD8* mRNA could be used as a prognostic marker candidate for salvage surgery after a local recurrence after definitive CRT.

In particular, our results showed that the oligoclonality of TIL was statistically induced in OSCC and ESCC with or without preceding CRT, when compared to the peripheral blood in healthy volunteers (Figure 2). This suggests that the clonal expansion of TIL could be induced by the stimulation of neoantigens derived from tumor cells through the process consisting of cancer antigen presentation, the priming and trafficking of T cells, and infiltration of T cells into tumors, which is commonly known as “the cancer-immunity cycle” reviewed by Chen and Mellman [30]. We also investigated whether the oligoclonality of TIL was more significantly induced after definitive CRT, because radiation and chemotherapy have been previously reported to induce neoantigens by disrupting cancer cells [17,31,32]. Preferably, we could compare the clonality of TIL between stage-adjusted ESCCs with and without preceding definitive CRT directly; however, it is difficult to recruit patients with stage II or III ESCCs that did not have preceding neoadjuvant therapies in Japan. This is because neoadjuvant chemotherapies or chemoradiation therapies are recommended before esophagectomy, which is supported by robust data from clinical studies [24,25]. Only two patients with stage II or stage III ESCCs that did not have preceding neoadjuvant therapies were enrolled in this study; accordingly, we predicted that the number of enrolled patients might not be enough to meet a statistical significance. Instead, we enrolled patients with stage II or III OSCCs in the study, because they likely had similar TMEs to ESCCs, and they shared the same pathologic diagnoses and carcinogenic pathways as described above [26,27,28]. Besides, patients with stage II or III OSCCs are usually treated by surgery alone, without any preceding neoadjuvant therapies. In fact, 13 patients with OSCCs that were treated by surgery alone were enrolled in this study (Table 1). As expected, diversity analyses of the TCR repertoire showed quite similar results in each diversity index between the OSCC and ESCC both treated by surgery alone (Figure 2), while significantly stronger oligoclonality in ESCC after definitive CRT than in OSCC was certainly observed (*p* < 0.001 in the Shannon-Weaver index H for *TRA*, *p* < 0.001 in Pielou’s evenness for *TRA*, *p* < 0.001 in the Shannon-Weaver index H for *TRB*, *p* < 0.001 in Pielou’s evenness for *TRB*, and *p* = 0.014 in inverse Simpson (1/λ) for *TRB*). Taken together, it might be plausible that the oligoclonality of TIL in ESCC after CRT might be more significantly induced than in ESCC treated by surgery alone. This clonal expansion of TIL induced by CRT is consistent with a previous study showing CD8+ TIL induction in ESCC after neoadjuvant chemotherapy, as evaluated by CD8 immunohistochemistry [33].

Next, we compared the diversity of the TCR repertoire between patients with good or poor prognoses after salvage surgery for ESCC that locally relapsed after definitive CRT. From previous reports, we predicted that patients with a good prognosis might contain a stronger clonal expansion of TIL, which might play an important role in improving the prognosis after definitive CRT [32,34]. As for TIL induction within tumors, the presence of CD8+ T-cell infiltration was previously reported to be a favorable prognostic marker in ESCC treated by surgery [35]. Moreover, higher post-CRT CD8+ TIL density was reportedly a favorable prognostic indicator after concurrent chemoradiation therapy for lung cancers [21]. Tumor remission and TILs during CRT were previously suggested to be predictive markers for pathological CR (pCR) in ESCC after CRT [23]. However, the results obtained in this study indicated no statistical differences between these two groups (Figure 3A,B). It could be possible that ESCCs remaining in pCR after definitive CRT could have a stronger expansion of clonality of TIL than the locally relapsed cases that were examined in this study. However, it is quite difficult to analyze the clonality of TIL by the TCR repertoire diversity only from the endoscopic biopsies, and physicians or surgeons usually watch carefully the patients with ESCCs after CRT without surgical treatments as far as they remain in clinical CR. The salvage surgery is only required at a local relapse after CRT. Therefore, our results just suggest that the clonal expansion of TIL is significantly induced after CRT, which is not statistically associated with the prognosis after salvage esophagectomy in cases with relapsed ESCCs, nor in cases remaining in pCR.

We then analyzed the TME to find the key factors separating the two groups, with a good prognosis and with a poor prognosis, after salvage surgeries. In the qPCR assay, we observed that the ratio of *PD-L1* mRNA to *CD8B* mRNA was statistically higher in those with a poor prognosis (*p* = 0.007), although both *PD-L1* and *CD8B*, adjusted by *GAPDH*, were not statistically different (Figure 4). These results prompted us to hypothesize that the clonal expansion of TIL, which occurred in all the cases after definitive CRT examined in this study, might not be enough and should be followed or accompanied by PD-L1 suppression, although the upregulation of PD-L1 is likely to be induced after neoadjuvant chemotherapy, as previously reported [33]. This hypothesis is plausible, because similar results have been reported in various human cancers; the lack of PD-L1 expression and the combined presence of CD8+ TIL were associated with favorable survival rates in stage III non-small cell lung carcinoma after CRT, as well as in patients with extrahepatic bile duct cancer after surgery and adjuvant chemoradiation therapy [18,22,36]; the immunophenotypes determined by TIL density and their spatial organization in the tumor, which might be influenced by the infiltration of CD8+ TIL and expression of PD-L1 or PD-1, were reportedly prognostic marker candidates of tongue squamous cell carcinoma [37]. The salvage surgery for ESCC that locally relapses after definitive CRT is surgically feasible; however, severe comorbidities such as pneumonia or anastomotic leakage are often reported [7,38]. Therefore, it is necessary to treat them with the optimal medical treatments; from this study, salvage surgery should be considered for ESCC with a low ratio of PD-L1/CD8 after definitive CRT, while a PD-1/PD-L1 blockade should be recommended for those with a high ratio of PD-L1/CD8 [30]. However, the results obtained in this study are based on the surgical specimens, not on the endoscopic biopsies or peripheral bloods. In order to establish the predictive marker, the results should be validated by another cohort by using endoscopic biopsies before the initiation of each treatment—in particular, the ratios of PD-L1/CD8 evaluated on the endoscopic biopsies before salvage esophagectomy.

## 4. Materials and Methods

### 4.1. Patients

The clinical and pathologic data of the enrolled patients are summarized in Table 1. All the tumors of seven ESCCs and 13 OSCCs were diagnosed as squamous cell carcinomas by board-certified pathologists. Among the seven ESCCs, two patients were treated only by surgery, and the others were primarily treated by definitive CRT, followed by salvage esophagectomy. Among the five patients treated by salvage esophagectomy, two survived without cancer recurrence, while the others died from esophageal cancer recurrence.

Specimens of esophageal cancer tissues were obtained from the enrolled patients with advanced ESCC who underwent surgical treatments with or without the preceding definitive CRT at Tohoku University Hospital. Furthermore, samples from patients with OSCCs were obtained from the those that underwent surgical treatments without any preceding neoadjuvant therapies at Tsurumi University Hospital or Kanto Rosai Hospital. This study was approved by the ethics committees of Tohoku University Graduate School of Medicine (permission number 2016-1-331), National Hospital Organization Sagamihara National Hospital (permission number 2018-051), Tsurumi University (permission number 1608), and Kanto Rosai Hospital (permission number 2018-25). Written informed consent was obtained prior to surgical treatment.

### 4.2. RNA Extraction from Tumor Tissues

Fresh specimens of ESCC tissues and OSCC tissues were obtained from all patients and immediately soaked in RNAlater Stabilization Solution (Invitrogen, Waltham, MA, USA) and/or eventually frozen at −80 °C. Total RNA was extracted from each specimen using the RNeasy Lipid Tissue Mini Kit (Qiagen, Venio, the Netherlands) according to the manufacturer’s instructions. Complementary DNA (cDNA) was synthesized from DNA-free RNA using the PrimeScript RT reagent Kit (Takara Bio, Tokyo, Japan) according to the manufacturer’s instructions.

### 4.3. T-Cell Receptor Repertoire Analysis

TCR Repertoire analysis was performed using the Repertoire Analysis Kit according to the manufacturer’s instructions (Repertoire Genesis, Ibaraki, Osaka, Japan). In brief, cDNA was synthesized using *hTRA* or *hTRB*-specific reverse transcription primers, which were subsequently conjugated with universal adapters. The first PCR was performed by unbiased PCR by using a nested PCR primer set that amplified the V, J, and C regions of *hTRA* and the V, D, J, and C regions of *hTRB*. The first PCR products were confirmed by electrophoresis. The obtained PCR products were purified using Agencourt AMPure XP (Beckman Coulter, Brea, CA, USA). The concentration of the purified product was measured using a Qubit Fluorometer (Thermo Fisher Scientific, Waltham, MA, USA) and, finally, diluted to 4 nM. The final products were sequenced by Mi-Seq (Illumina, San Diego, CA, USA). The obtained sequence data were analyzed by TCR repertoire analysis software developed by Repertoire Genesis (Repertoire Genesis, Ibaraki, Osaka, Japan).

As a normal control, RNA was extracted from peripheral white blood cells from 18 healthy volunteers enrolled in this study under permission number 2015121708 of the ethics committee of the Clinical Research Center for Rheumatology and Allergy, National Hospital Organization Sagamihara National Hospital, as previously described [39]. TCR repertoire analyses were performed as described above.

### 4.4. cDNA Synthesis and Quantitative Polymerase Chain Reaction

The expression levels of immune response-related genes, including T-cell-related clusters of differentiation antigens and biomarkers, were measured by quantitative polymerase chain reaction (qPCR) using the Bio-Rad CFX96 system (Bio-Rad, Hercules, CA, USA). PCR primers for *CD8B* and *GAPDH* were purchased from Takara Bio Inc. (Shiga, Japan), and the primers for *PD-L1* were obtained from a previous report [40]. The sequences of the primers were as follows:*CD8B*-forward 5′-GGCATCTACTTCTGCATGATCGTC-3′*CD8B*-reverse 5′-TGGGTAACCGGCACACTCTC-3′*PD-L1*-forward 5′-AAATGGAACCTGGCGAAAGC-3′*PD-L1*-reverse 5′-GATGAGCCCCTCAGGCATTT-3′*GAPDH*-forward 5′-GCACCGTCAAGGCTGAGAAC-3′*GAPDH*-reverse 5′-ATGGTGGTGAAGACGCCAGT-3′.

cDNAs were synthesized from 500 ng of total RNA isolated from each specimen of ESCC or OSCC using the Prime Script RT reagent Kit (Takara, Shiga, Japan) for each of the 10 qPCR reactions. The PCR mixture consisted of 5 μL of SsoFast EvaGreen Supermix (Bio-Rad, Hercules, CA, USA), 3.5 μL of RNase/ DNase-free water, 0.5 μL of 5-μM primer mix, and 1-μL cDNA in a final total volume of 10 μL. This was then applied to the CFX96 system. Cycling conditions were as follows: 30 s at 95 °C, followed by 50 cycles of 1 s at 95 °C and 5 s at 60 °C. At the end of each program, a melting curve analysis was performed from 70 °C to 94 °C to confirm homogeneity of the PCR products. All assays were performed in triplicate, and mean values were used to calculate the gene expression levels. For calibration of the absolute quantification, a standard curve for each target was prepared by five 10-fold serial dilutions of the standard nucleotide that was generated by the linearized plasmid cloned with the PCR product of each target. This was confirmed by DNA sequencing using the Sanger method and was quantified by SmartSpec3000 (Bio-Rad, Hercules, CA, USA).

### 4.5. Diversity Metric Calculation

The clonality index was defined as either Pielou’s evenness, the Shannon-Weaver index H, or inverse Simpson’s index (1/λ) and calculated as described previously [39,41].

### 4.6. Statistical Analysis

For continuous variables, the Shapiro-Wilk *W* test for normality was performed for each group, at first. If the parametric conditions were fulfilled, an unpaired Student’s *t*-test was performed, except for those that had statistical significance (*p* < 0.05) for the Levene’s test, which were instead analyzed by Welch’s *t*-test. For nonparametric conditions, the Mann-Whitney U test was performed. All analyses were performed using SPSS version 24 (IBM, Armonk, NY, USA). A bilateral *p*-value below 0.05 was considered statistically significant.

## 5. Conclusions

In this study, we showed a significant induction of oligoclonality of TIL in ESCC or OSCC when compared to healthy volunteers; further, we suggested the clonal expansion of TIL might be more notably induced when preceded by definitive CRT. Furthermore, our results suggested that a low ratio of *PD-L1* mRNA/*CD8B* mRNA in tumor tissues after definitive CRT could be a potential prognostic maker for salvage surgery, although the clonal expansion of TIL itself was not enough to improve the prognosis there. We predicted that not only the clonal expansion of TIL but, also, *PD-L1* suppression, which is likely upregulated during chemotherapy and chemoradiation therapy, should play an important role in improving the prognosis. This viewpoint was supported by previous results in other types of cancers, such as extrahepatic bile duct cancers, non-small cell lung carcinoma, or tongue squamous cell carcinoma [18,21,22,36,37]. However, the results presented here should be validated by another clinical cohort because of the limited number of enrolled patients, especially for the prognostic study of salvage surgery after definitive CRT, and also validated by using endoscopic biopsies before salvage surgeries to establish the predictive markers.

## Figures and Tables

**Figure 1 ijms-22-01098-f001:**
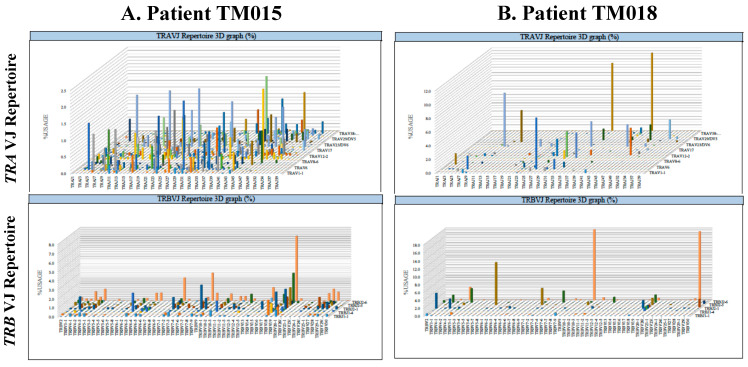
The typical results of the *T-cell receptor α* (*TRA*) and *TCR B* (*TRB*) repertoires, each of esophageal squamous cell carcinoma (ESCC) treated by surgery only (**A**) and ESCC relapsed after definitive chemoradiation therapy (CRT) (**B**). The *TRA* repertoire and *TRB* repertoire are shown in the upper and lower panels, respectively. Three-dimensional graphs indicate the combination of using *TRAJ* on the x-axis and *TRAV* on the y-axis or of *TRBV* on the x-axis and *TRBJ* on the y-axis, with the frequency (percentage) of each clone on the z-axis. Oligoclonal expansion is observed more in ESCC relapsed after definitive CRT (**B**) than in ESCC treated by surgery alone (**A**).

**Figure 2 ijms-22-01098-f002:**
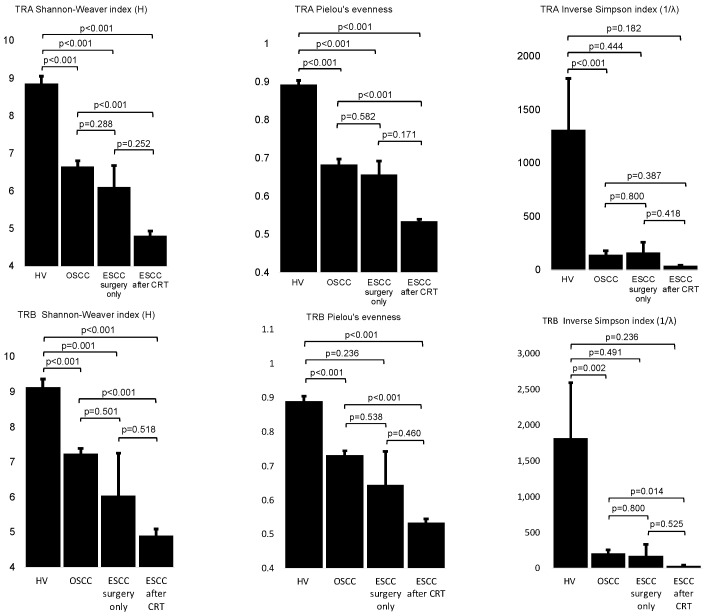
T-cell receptor (TCR) diversity indices in healthy volunteers (HV), oral squamous cell carcinoma (OSCC) treated by surgery alone, and esophageal squamous cell carcinoma (ESCC) with or without preceding chemoradiation therapy. Each bar indicates a mean value of each TCR repertoire index with an error bar as the standard error.

**Figure 3 ijms-22-01098-f003:**
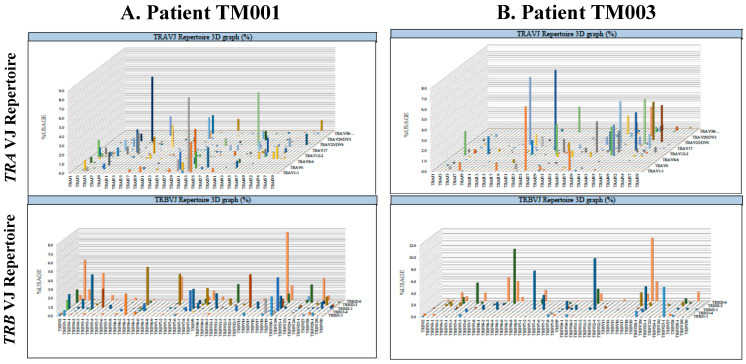
The typical results of the *TRA* and *TRB* repertoires in esophageal squamous cell carcinomas (ESCCs).

**Figure 4 ijms-22-01098-f004:**
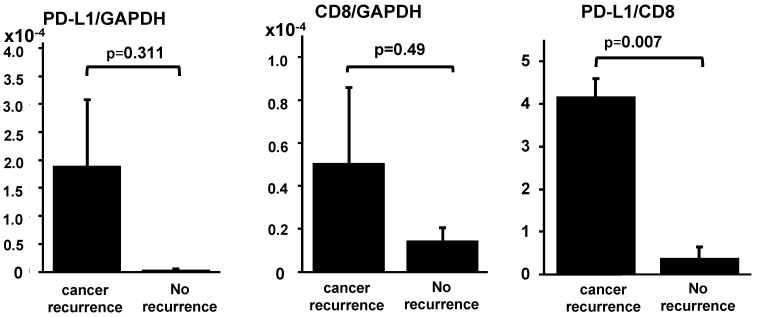
The results of tumor microenvironment analyses by quantitative polymerase chain reaction in both the cancer recurrence group and no recurrence group after salvage esophagectomy for local failure of definitive CRT. Each bar indicates a mean value with an error bar as the standard error. The ratio of *PD-L1* mRNA to *CD8B* mRNA was statistically different between patients with good prognoses and with poor prognoses after the salvage surgeries (*p* = 0.007). However, each expression level of *PD-L1* and *CD8B*, adjusted by *GAPDH* mRNA, was not statistically different between these two groups.

**Table 1 ijms-22-01098-t001:** The clinical and pathologic data of patients with esophageal squamous cell carcinoma (ESCC) or with oral squamous cell carcinoma (OSCC) examined in this study. CRT: chemoradiation therapy.

Disease	First-Line Therapy	Radiation Dose (gy)	Concurrent Chemotherapy	fTNM	Prognosis After Salvage Esophagectomy		
					Cancer Recurrence	Prognosis	Recurrence-Free Survival (Months)
ESCC	CRT	60	CDDP+5-FU	T2N1M0	yes	cancer death	18.4
ESCC	CRT	60	CDDP+5-FU	T2N1M0	no	survive	22.9
ESCC	CRT	60	CDDP+5-FU	T3N0M0	no	survive	74.1
ESCC	CRT	60	CDDP+5-FU	T4N1M0	yes	cancer death	6.6
ESCC	surgery alone	0	no	T2N1M0	no	survive	117.7
ESCC	surgery alone	0	no	T3N0M0	no	died from other disease	67.2
ESCC	CRT	60	CDDP+5-FU	T3N0M0	yes	cancer death	9.5
OSCC	surgery alone	0	no	T2N0M0			
OSCC	surgery alone	0	no	T1N1M0			
OSCC	surgery alone	0	no	T2N2bM0			
OSCC	surgery alone	0	no	T2N1M0			
OSCC	surgery alone	0	no	T4aN0M0			
OSCC	surgery alone	0	no	T4aN3bM1			
OSCC	surgery alone	0	no	T1N1M0			
OSCC	surgery alone	0	no	T4N1M0			
OSCC	surgery alone	0	no	T1N1M0			
OSCC	surgery alone	0	no	T3N0M0			
OSCC	surgery alone	0	no	T1N1M0			
OSCC	surgery alone	0	no	T2N2bM0			
OSCC	surgery alone	0	no	T3N0M0			

## Data Availability

The data presented in this study are available on request from the corresponding author.

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
