# Peer review of "Clonal Expansion of Tumor-Infiltrating T Cells and Analysis of the Tumor Microenvironment within Esophageal Squamous Cell Carcinoma Relapsed after Definitive Chemoradiation Therapy"

_ijms, 2021, doi:10.3390/ijms22031098_

Round 1
Reviewer 1 Report
This is an important study exploring the prognostic features within tumor microenvironments in patients who need salvage esophagectomy after definitive CCRT for esophageal SqCC, given that no other therapeutic options than salvage surgery, which tends to be potentially very morbid, are left for these patients who developed local recurrence after maximal radiotherapy. They found a clear indicator, the PD-L1 mRNA to CD8B mRNA in TME, that could enable us to make a tough decision on whether to do a salvage surgery, although their findings need to be validated through other study samples.
In spite of this important finding, what the authors addressed the background of this study, the hypothesis, and the objectives in the Introdcution section seems to be vague and not clearcut. The strengths of their findings should be focused on the features that could differentiate the prognosis of patients undergoing salvage esophagectomy based on clonal expansion of TIL and analysis of TME. This aspect as well as the obejctives and hypothesis should be more clearly described in the Introduction section.
All the materials for analysis are surgical specimens, not endoscopic specimens. Although there might be substantial difference in TME natures between surgical and endoscopic specimens, we cannot help but rely on endoscopic specimens before we have to decide on salvage esophagectomy, because surgical specimens are not available at that time point. This limitation should be discussed further in the Discussion section.
Author Response
We appreciate the time and effort you and each of the reviewers have dedicated to providing insightful feedback on ways to strengthen our paper. Thus, it is of great pleasure that we resubmit our article for further consideration. We have incorporated changes that reflect the detailed suggestions you have graciously provided. We also hope that our edits and the responses we provide below satisfactorily address all the issues and concerns you and the reviewers have noted.
To facilitate your review of our revisions, the following is a point-by-point response to the questions and comments delivered in your letter dated January 4th, 2021.
“The strengths of their findings should be focused on the features that could differentiate the prognosis of patients undergoing salvage esophagectomy based on clonal expansion of TIL and analysis of TME. This aspect as well as the obejctives and hypothesis should be more clearly described in the Introduction section.”
We have reflected the above comment by adding sentences in the introduction section ( lines 66-67 and lines 79-80) to establish a clearer focus.
We have replaced [demonstrates] with [suggests] to use more precise terms (line 69).
“Although there might be substantial difference in TME natures between surgical and endoscopic specimens, we cannot help but rely on endoscopic specimens before we have to decide on salvage esophagectomy, because surgical specimens are not available at that time point. This limitation should be discussed further in the Discussion section.”
We have reflected the above comment by adding sentences in the discussion section ( lines 248-252) and in the conclusion section (lines 342-343).
Again, thank you for giving us the opportunity to strengthen our manuscript with your valuable comments and queries. We have worked hard to incorporate your feedback and hope that these revisions persuade you to accept our submission.
On behalf of the authors,
Takahiro Mori, M.D. Ph.D.
Reviewer 2 Report
This is a nice piece of work investigating the role of tumor-infiltrating T cells (TILs) in cases of recurrent esophageal squamous cell carcinoma (ESCC) after chemoradiotherapy. In particular, the authors conducted diversity analyses of the T-cell receptor repertoire of TIL and demonstrate the molecular features of the tumor microenvironment (TME) within post-chemoradiotherapy ESCC cases. The results were interesting, revealing the presence of specific immunological features initiated by chemoradiotherapy. Moreover, the ratio of PD-L1 mRNA to CD8B mRNA in TME was significantly associated with poor prognosis after salvage surgery.
The techniques used were appropriate and described with plenty details. Overall, this is a well-designed study with rigorous methods. The discussion is well-balanced, and the statements are supported by the data. The work is interesting because gain initial insight into the possible prognostic role of these markers in recurrent ESCC. There are few minor concerns to revise that are described below:
- Materials and methods: please add a brief paragraph reporting the number and the clinicopathological data of the patients included in this study.
- Table: I suggest simplifying the table by deleting the columns 1 (“patient ID”), 7 (“pathology”), and 11 (“prognosis group after salvage surgery”).
- Discussion: The authors discuss the phenomenon of TME and the role of TILs. As the importance of the topic, I recommend adding some considerations related to the role of tumor-infiltrating T cells, as described by Chen & Mellmann. In particular, I suggest to briefly discuss the role of immune-phenotype in epithelial tumors (for your convenience: doi: 10.1002/cam4.3440).
Author Response
We appreciate the time and effort you and each of the reviewers have dedicated to providing insightful feedback on ways to strengthen our paper. Thus, it is of great pleasure that we resubmit our article for further consideration. We have incorporated changes that reflect the detailed suggestions you have graciously provided. We also hope that our edits and the responses we provide below satisfactorily address all the issues and concerns you and the reviewers have noted.
To facilitate your review of our revisions, the following is a point-by-point response to the questions and comments delivered in your letter dated January 4th, 2021.
“Materials and methods: please add a brief paragraph reporting the number and the clinicopathological data of the patients included in this study.”
We have reflected the above comment by creating “Patients” section, in which we have briefly described the number and clinicopathological data of the patients included in this study in the Materials and Methods section ( lines 256-261).
“Table: I suggest simplifying the table by deleting the columns 1 (“patient ID”), 7 (“pathology”), and 11 (“prognosis group after salvage surgery”).”
We have corrected Table 1 as suggested, by deleting the columns 1 (patient ID), 7 (pathology) and 11 (prognosis group after salvage surgery).
“Discussion: The authors discuss the phenomenon of TME and the role of TILs. As the importance of the topic, I recommend adding some considerations related to the role of tumor-infiltrating T cells, as described by Chen & Mellmann. In particular, I suggest to briefly discuss the role of immune-phenotype in epithelial tumors (for your convenience: doi: 10.1002/cam4.3440).”
We have reflected the above comments by correcting sentences by referring the review by Chen and Mellmann (lines 185-187), and by adding description of the immune-phenotypes reported by Troiano et al. (lines 240-243 and line 340)
Again, thank you for giving us the opportunity to strengthen our manuscript with your valuable comments and queries. We have worked hard to incorporate your feedback and hope that these revisions persuade you to accept our submission.
On behalf of the authors,
Takahiro Mori, M.D. Ph.D.